# Functional Characteristics and Regulated Expression of Alternatively Spliced Tissue Factor: An Update

**DOI:** 10.3390/cancers13184652

**Published:** 2021-09-16

**Authors:** Kateryna Matiash, Clayton S. Lewis, Vladimir Y. Bogdanov

**Affiliations:** Division of Hematology/Oncology, Department of Internal Medicine, University of Cincinnati College of Medicine, Cincinnati, OH 45267, USA; matiaska@mail.uc.edu (K.M.); lewis2c3@ucmail.uc.edu (C.S.L.)

**Keywords:** tissue factor, alternative splicing, integrins, pancreatic ductal adenocarcinoma, breast cancer, biomarker, anti-cancer biologics

## Abstract

**Simple Summary:**

Alternatively spliced tissue factor (asTF) is a naturally occurring isoform of tissue factor (TF) generated via the omission of exon 5 during the processing of TF’s primary transcript. In human and mouse, asTF protein features a unique C-terminus that lacks a transmembrane domain, rendering it soluble. asTF protein is able to associate with a subset on integrins on cell surfaces, which can trigger outside-in signaling programs in a variety of cell types. In this review, we discuss recent findings on asTF’s proto-oncogenic effects, regulatory mechanisms enabling asTF’s biosynthesis, and asTF’s potential as a biomarker and therapeutic target.

**Abstract:**

In human and mouse, alternative splicing of tissue factor’s primary transcript yields two mRNA species: one features all six TF exons and encodes full-length tissue factor (flTF), and the other lacks exon 5 and encodes alternatively spliced tissue factor (asTF). flTF, which is oftentimes referred to as “TF”, is an integral membrane glycoprotein due to the presence of an alpha-helical domain in its C-terminus, while asTF is soluble due to the frameshift resulting from the joining of exon 4 directly to exon 6. In this review, we focus on asTF—the more recently discovered isoform of TF that appears to significantly contribute to the pathobiology of several solid malignancies. There is currently a consensus in the field that asTF, while dispensable to normal hemostasis, can activate a subset of integrins on benign and malignant cells and promote outside-in signaling eliciting angiogenesis; cancer cell proliferation, migration, and invasion; and monocyte recruitment. We provide a general overview of the pioneering, as well as more recent, asTF research; discuss the current concepts of how asTF contributes to cancer progression; and open a conversation about the emerging utility of asTF as a biomarker and a therapeutic target.

## 1. Introduction

It is remarkable that, after 30+ years following its cloning and initial characterization [1,2], tissue factor (TF) remains an intensely studied molecule, largely because there is still a lot to be learned in the realm of TF’s contribution to the pathobiology of various disease states—most notably, vascular and neoplastic pathologies. Throughout the decades, TF has acquired multiple aliases, e.g., thromboplastin, tissue factor, CD142, and “coagulation factor III”, which was used by Human Genome Organization to set the gene name “*F3*” (analogously, the name of the murine gene is *cf3*). As recently as this year, vascular TF was reported, for the first time, to contribute to trimethylamine N-oxide (TMAO) induced thrombosis, whereas co-expression of TF and podoplanin by glioblastoma cells was discovered to cooperatively impact tumor microthrombosis [3,4]; an emerging role of TF in COVID-19-associated coagulopathy is covered in the timely review by Mackman and colleagues [5]. Here, we focus on the form of TF that remains the only biochemically soluble TF form known to occur naturally, via alternative processing of TF’s primary transcript. This form, termed “alternatively spliced TF” (asTF), should not be confused with “soluble TF”, which stands for an engineered ectodomain of the classical full-length TF form (flTF), exemplified by the elegant recent work of Sorensen et al., Vadivel et al., and Jin et al. [6,7,8]. In 2015, we covered the issues associated with the misuse of “soluble TF” nomenclature at some length [9]; a more recent example of “soluble TF” research, wherein the nature of TF being studied is uncertain, comprises the report by Suehiro and colleagues [10]. In this review, we refer to TF protein as “TF” or “total TF”; “flTF” and “asTF” to specify mRNA and/or protein TF isoforms; and *F3*/*cf3* in reference to the human or murine TF genes, respectively (see Figure 1 for details). We perceive asTF to be an inherent yet discreet component of the “TF system”; asTF’s unique biochemical structure and functional characteristics allow the possibility to detect and target asTF selectively. Here, we provide an overview of recent data pertaining to asTF’s expression in various tissues; asTF’s mechanisms of action and the consequences of its biosynthesis by various cell types in certain tissue compartments; and asTF’s emerging potential as a circulating biomarker and a therapeutic target.

## 2. asTF Is Overexpressed in Multiple Malignancies

As recently reported by Unruh and Horbinski, Pan-Cancer TCGA analysis shows that high *F3* expression in neoplastic tissue significantly correlates with worse overall survival [13]. Further analysis of TCGA survival data reveals that renal and pancreatic cancer patients specifically have decidedly unfavorable survival prognoses when intratumoral *F3* expression is high (*p* < 0.001) [14,15]. Intriguingly, transcriptomic analysis of global splicing events in pancreatic ductal adenocarcinoma (PDAC) specimens revealed that the most common alternative splicing event detected was exon skipping—the same type of event that allows for the biosynthesis of asTF [16]. TF is highly expressed in the leading edge of tumors; circulates in the blood of cancer patients; contributes to malignancy-associated thrombosis; and can promote cancer cell proliferation and metastasis through non-hemostatic signaling [17,18,19,20]. Hisada and Mackman, as well as Unruh and Horbinski, provide an excellent overview of TF overexpression in various cancer types, as well as of the mechanisms that contribute to it [13,21]. Before the discovery and characterization of asTF in 2003, however, it had been impossible to differentiate between asTF and flTF expression via immunohistochemical staining, so the majority of the studies were unable to differentiate between isoforms and described “total TF” overexpression. More recently, studies profiling TF expression in lung, cervical, pancreatic, and colon cancers showed that both asTF and flTF were overexpressed in cancer lesions and cell lines [22,23,24,25].

Originally discovered in HL-60 cells (a human promyelocytic cell line derived from a patient with acute leukemia), asTF was later shown to also exist in the mouse, where it was found to be co-expressed with flTF in multiple organs, such as spleen, liver, heart, kidneys, and brain, and it also localized to spontaneously formed thrombi [11,26]. Profiling of human pancreatic cancer cell lines for TF expression via conventional RT-PCR by Haas and colleagues revealed a high expression of flTF (8/8 cell lines) and concomitant asTF expression (6/8 cell lines) [22]. Two years later, another group published a study on the assessment of flTF and asTF mRNA expression via qRT-PCR in non-small cell lung cancer (NSCLC) patients; their results revealed that, while flTF was present in healthy lungs and overexpressed in pulmonary tumors, asTF expression was confined exclusively to malignant tissue samples [27]. Goldin-Lang and colleagues found that asTF mRNA is an independent prognostic marker in NSCLC, but they also noted that asTF transcripts constituted less than 1 percent of all TF transcripts [27]. Interestingly, they found no correlation between flTF and asTF mRNA expression levels in NSCLC patients [27]. In 2013, Kocaturk et al. analyzed flTF and asTF protein expression by immunohistochemistry (IHC) staining in 574 breast cancer (BC) tissue specimens and found a distinct association between asTF expression and tumor size. Both asTF and flTF correlated to tumor grade, and asTF was shown to promote BC cell spread and proliferation via integrin ligation [28]. Neither asTF nor flTF primary tumor expression correlated with tumor cell spread to lymph nodes in BC patients [28]. Around the same time, our lab reported that, together with mouse flTF, mouse asTF protein was highly expressed in genetically engineered mouse models (GEMMs) of pancreatic cancer but not in the normal pancreas (except for pancreatic islets), as assessed by IHC [29]. asTF’s expression in murine pancreatic islets echoed the earlier observations of Beuneu and colleagues in human islet cells [30]. We also showed that human asTF was overexpressed in PDAC lesions, and its protein levels positively correlated with CD68+ cell staining, indicating an increased macrophage infiltration in tumors with high asTF expression [31]. Circulating levels of asTF, measured by our enzyme-linked immunosorbent assay (ELISA) in the plasma of PDAC patients, when compared to healthy age-and-sex-matched healthy individuals, were also significantly higher [32]. Moreover, PDAC patients with high levels of circulating asTF were less likely to qualify for tumor resection [32]. However, similar to the BC tissue array results, circulating asTF levels in PDAC did not correlate with nodal metastasis (pN) [28,32]. We note, in this regard, that a study validating the most recent tumor–node–metastasis protocol for PDAC patients revealed that local tumor size (pT), rather than pN, is prognostically relevant for patient outcomes [33].

Wu and colleagues showed that asTF and flTF transcript levels measured via conventional RT-PCR were significantly more abundant in gastric adenocarcinoma compared to adjacent normal tissue [24]. None of the clinical parameters correlated with asTF and/or flTF expression in gastric cancer patients, but univariate and multivariate Cox modeling showed that either asTF or flTF mRNA expression can serve as an independent prognostic risk factor for gastric cancer [24]. Clinical implications of these particular analyses remain uncertain because measuring asTF and flTF mRNA in gastric tumor tissue is not very practicable. Nonetheless, these findings will hopefully prompt further inquiries into asTF protein levels in, and its possible role as a driver of this malignancy. Another group reported that total TF knockdown can inhibit gastric cancer cell proliferation and spread, but the contribution of each TF isoform individually to the observed phenotype was not addressed [34]. In 2019, the qRT-PCR assessment of asTF and flTF transcript expression by Pan et al. extended the inquiry into the two TF isoforms’ presence in multiple cancer tissues and cell lines [25]. The researchers found that cervical, breast, colorectal cancer, and hepatoblastoma cell lines expressed asTF and flTF, with all cell lines expressing at least 200 times more flTF transcripts than asTF transcripts [25]. Additionally, Pan et al. reported that lung, esophageal, cervical, and breast cancer resection specimens displayed the same pattern of TF isoform expression as the cell lines they had screened [25]. Pan and colleagues went on to evaluate the effects of chenodeoxycholic acid (CDCA), a bile acid, and apolipoprotein M on TF isoform expression, finding that CDCA enhanced flTF, but not asTF, expression in hepatoblastoma HepG2 cells; however, CDCA promoted asTF expression in umbilical vein endothelial cells [25]. They also found that, in colorectal cancer, cell line Caco-2 asTF expression was upregulated in response to ApoM overexpression. [25]. This study, however, does not provide an insight into the absolute levels of each mRNA isoform expressed, as the reported asTF and flTF levels were reported as a percentage of asTF levels [25]. Furthermore, without an evaluation of protein levels of the two biochemically distinct entities—a membrane-bound flTF and a soluble, secreted asTF—it is difficult to make inferences about the relative contribution of the two TF isoforms to the investigated disease settings.

Cumulatively, these recent studies show that both asTF and flTF are highly expressed in malignant cells and tissues and therefore should both be considered when evaluating the TF system’s contribution to neoplastic pathobiology, particularly in gastrointestinal cancers.

## 3. Consequences of asTF’s Interaction with Integrins on Surfaces of Benign and Malignant Cells

One of the most recent studies on TF in cancer—specifically, pancreatic cancer—by the Flick laboratory reported that TF promotes PDAC progression through thrombin generation and the subsequent activation of protease-activate receptor 1 (PAR-1), a member of a G-protein coupled receptor subfamily [17]. Most of the canonical flTF signaling tied to cancer progression is thought to be initiated after flTF activates FVII and forms a flTF/FVIIa complex, leading to the cleavage of PAR-2 [21]. However, similar to asTF, the flTF/FVIIa complex can also interact with cell-surface integrins, initiating pro-survival signaling cascades [35]. For example, a recent article by Åberg and colleagues reports that the flTF/FVIIa complex can induce calveolin-1 phosphorylation and reduce apoptosis via β1 integrin complex activation and cross-talk with insulin growth factor receptor 1 (IGF-R1) in breast and prostate cancer cell lines [36]. Interestingly, PARs themselves have been implicated in influencing integrin activation and signaling. For instance, PAR-1 and PAR-2 were both shown to promote adhesion of fibroblasts to fibronectin via α5β1 receptor after activating PI3K and Src kinases, causing this integrin dimer to transition into a high-affinity conformation and associate with talin and kindlin, which promotes the assembly of an adhesome and involves pro-migratory and pro-proliferative signaling via PI3K and Src [37]. Another example is a study showing that PAR-3-deficient pancreatic cancer cell line PANC-1 overexpresses E-cadherin and integrin αv, which results in the increased cell adhesion and may reflect the state of increased re-attachment upon cancer cell colonization in premetastatic niches [38]. In this section, we overview how asTF can activate integrins and promote cancer cell spread, independent of protease co-factors and/or PARs (Figure 2).

The first study describing the effects of asTF on cell proliferation and migration was carried out using endothelial cells (ECs); it revealed that the pathway responsible for the phenotype was integrin-mediated [40]. The coating of the Boyden chamber inserts with human asTF promoted EC migration via integrin αvβ3, while non-directional capillary formation was induced through α6β1 [40]. asTF also enhanced ex vivo aortic ring sprouting, which was reduced by blocking both β1 and β3 integrins with respective blocking antibodies [40]. Four years later, Kocaturk and colleagues observed that human asTF-enhanced BC cell proliferation was also enacted via integrin α6β1 [28]. Shortly thereafter, human asTF was shown to promote BC cell migration and in vivo growth synergistically with estradiol in an ER+ setting [42]. More recent studies from our group have been focused on how asTF potentiates cancer cell proliferation and spread in PDAC—one of the deadliest, least treatable, and most difficult to diagnose malignancies.

PDAC patients are confronted with a lack of early detection strategies, and once the disease has progressed to stages III–IV, it is nearly impossible to cure [43]. PDAC meets the definition of a recalcitrant cancer: 5 year survival of patients after diagnosis has only recently reached 10% in the United States, with over 40,000 people succumbing to the disease annually; worldwide 5 year survival rates are ever more somber at only 6% [43,44,45]. As noted earlier, asTF is overexpressed in PDAC lesions to different degrees, with the largest tumors staining heavily (mostly within malignant pseudo-ductal cells) with a specific anti-asTF antibody [31]. Hobbs and colleagues observed that overexpression of asTF in TF-null MiaPaCa-2 PDAC cells promoted their proliferation and subcutaneous tumor neovascularization [46]. Subsequently, our group demonstrated that constitutive asTF overexpression in a human grade III PDAC cell line, Pt45.P1, promoted cancer cell spread and tumor neovascularization in an orthotopic mouse model [31]. asTF overexpression in Pt45.P1 did not influence tumor size, likely due to a short end-point of the experiments (under 4 weeks post-implantation) [31]. RNA microarray analysis of Capan-1 and Pt45.P1 cells treated with recombinant asTF revealed an upregulation of genes tied to cell migration, p38 MAPK signaling, and wound healing pathways [31]. While studying asTF’s effects in malignancy, our laboratory also developed an inhibitory rabbit monoclonal antibody that targets the unique c-terminus of human asTF, termed RabMab1 (previously Rb1 [28,47]; US patent US 9,555,109 B2). Our first study describing hybridoma-derived RabMab1 showed that, both in vitro and in vivo, Pt45.P1 cells were more invasive, formed larger tumors, and metastasized after doxycycline-inducible overexpression of asTF [47]. Interestingly, in this study, asTF-enhanced PDAC cell migration was enabled by α6β1 integrins, while van der Berg et al. showed that EC migration was mainly potentiated by asTF via αvβ3 integrins, which could be due to integrin availability on the surface of different cells, or because of cancer cells’ propensity to employ unconventional signaling axes to their advantage [40,47]. Importantly, we found that Pt45.P1 cells orthotopically implanted into either TF-low or TF-heterozygous mice displayed similar rates of growth and levels of activated Akt, a major downstream target of asTF. However, systemic spread and macrophage infiltration of the tumor tissue were significantly more pronounced in TF-heterozygous mice, suggesting that host-derived TF isoforms significantly contribute to PDAC progression [47]. Our ongoing studies employing human and murine PDAC organoids, along with in vitro co-culture approaches, will help better elucidate whether asTF produced by PDAC cells and, possibly, stromal (“host”) cells impact the levels and/or phenotype of cancer-associated fibroblasts, as well as tumor-infiltrating leukocytes [48,49].

PDAC tumor tissue contains high levels of various fibrous proteins, such as collagen and laminin, leading to vessel collapse and oxygen/blood supply deprivation; this renders the PDAC tumor microenvironment (TME) highly hypoxic and acidic [48]. In collaboration with Dr. Weber’s laboratory, we found that subjecting asTF-overexpressing Pt45.P1 cells to hypoxia promoted the overexpression of carbonic anhydrase IX, a cell surface receptor that can help promote tumor acidosis by catalyzing the conversion of water and carbon dioxide to bicarbonate and protons [50,51,52]. These observations suggest that, in PDAC lesions with high intratumoral asTF expression, there is likely to be an increased acidification of the TME, which is thought to stimulate metastatic spread [50].

Thus far, asTF has been shown to contribute to various malignancy states when either added to cells in vitro in a recombinant form or being overexpressed in a constitutive or an inducible manner [28,31,40,47]. Having found that, in some cancer patients, asTF is present in the circulation in the nanomolar range, we routinely assess asTF protein’s effects on cellular processes within that range; however, potential issues with protein overexpression-based approaches must be considered [28]. For example, overexpression may cause stoichiometric imbalance within the cell, overloading protein turnover machinery by forcing it to prioritize the degradation of the overexpressed entity over others that are normally (to the extent that the term may apply to cancer cells) turned over proteins [53]. Such a stoichiometric push may lead to signaling defects, and the observed phenotype could be the result of these alterations to the levels of any number of proteins rather than the one of interest [53]. We did not see a decrease in asTF levels post RabMab1 treatment, but tumor growth and cell proliferation were visibly reduced, perhaps, in part, alleviating the concerns about turnover-altering overexpression of asTF in other settings. Another possible artifact of protein overexpression is the promiscuity of its interactions with other proteins in supra-physiological concentrations, forcing phenotypic changes that would, otherwise, not be seen, even in a malignant setting [53]. We have so far found that asTF interacts with integrins at a unique binding site at the stalk of the dimer’s extracellular portion, but other possible binding partners may exist for this TF isoform. We have been working on rendering a predicted asTF protein structure to map out possible protein binding sites. Recent breakthroughs in artificial intelligence research, such as the creation of a protein structure predictive software, termed AlphaFold 2.0, may help us succeed in modeling asTF’s 3D structure without the necessity to crystallize the protein [54]. Possible undesirable effects of protein overexpression on observed phenotypes may also be circumvented by downregulating asTF; however, targeted downregulation of this spliced TF isoform is difficult to achieve due to the ‘leaky’ nature of exon-specific shRNA-based targeting. In addition to using our human asTF-blocking antibody RabMab1, we are employing CRISPR-based splice site manipulation to alter exon 5 skipping rates to achieve a more physiologic, asTF-favoring change in “total TF” output [55]. Interestingly, it was observed that, when rendered to be more aggressive post in vivo passaging, a BC cell line MDA-MB-231-mfp expressed a significantly higher amount of asTF while the levels of flTF were unchanged; concomitantly, MDA-MB-231-mfp cells expressed higher levels of several splicing regulatory (SR) proteins that participate in TF pre-mRNA processing [28].

## 4. asTF Biosynthesis: Regulated vs. Aberrant Alternative Splicing

Pre-mRNA splicing is a process that involves the excision of non-coding regions called introns and the joining of coding regions (exons) into a mature or messenger RNA (mRNA), which is further processed and exported out of the nucleus into the cytoplasm to be translated into a protein. During alternative pre-mRNA splicing, certain exons may be skipped, resulting in the biosynthesis of structurally distinct proteins that help diversify the functions of a single gene. One of the most well-studied examples of homeostatic alternative splicing is the processing of tropomyosin pre-mRNA, which has preserved the need for mutually exclusive exon retention throughout eukaryotic evolution [56]. The generation of tissue and organ specific tropomyosin isoforms allows for the functional specialization of over 40 unique proteins that all arise from one gene [56]. Similar to many physiologic homeostatic processes, alternative splicing is employed by cancer cells as a means to generate proteins that can be effectively harnessed for enhanced cell proliferation, survival, and motility [57]. For example, metabolically active cancer cells switch from aerobic to anaerobic glycolysis by altering the incorporation pattern of mutually exclusive exons 9 and 10 in pyruvate kinase isoforms PKM1 and PKM2 [58]. c-Myc-driven upregulation of a subset of splicing regulatory factors results in the inclusion of exon 10 over exon 9, thereby promoting the expression of PKM2 and a switch to anaerobic glycolysis, which contributes to the Warburg effect [58,59]. Cancer cell reliance on glycolysis alone for ATP generation is thought to allow for the rapid production of energy that can be utilized for enhanced cell proliferation and protein biosynthesis, without utilizing extensive metabolic pathways maintained during homeostasis [59]. Similarly, aberrant alternative splicing influences all major processes involved in malignancy: anti-apoptotic forms of proteins are expressed to aid cancer cells in resisting cell death; cell-cycle-promoting protein isoforms allow for rapid cell division; and pro-angiogenic spliced variants are produced to augment tumor neovascularization [57].

To date, cancer-specific asTF overexpression has only been seen concomitant with flTF upregulation, implicating shared transcriptional drivers for the expression of both isoforms. For instance, our group recently showed that dual mammalian target of rapamycin complex (mTORC) inhibition decreased both flTF and asTF expression in human pancreatic neuroendocrine tumor cells via downregulation of the transcription factor Sp1 [60]. Post-transcriptional mechanisms of TF pre-mRNA splicing, while understudied in malignant settings, have been evaluated in monocytes and EC. In 2008, Tardos et al. assessed SR protein-mediated alternative pre-mRNA processing of TF in monocytes, using a splicing reporter construct for human TF that contained a genomic *F3* sequence starting with the 3′ end of exon 4 and extending into the 5′ end of exon 6 [61]. Site-directed mutagenesis of SR protein binding motifs in the exon 5 region of the reporter construct revealed that two splicing factors, SRSF1 and SRSF6, promote exon 5 inclusion and, by inference, the generation of flTF mRNA over that of asTF mRNA [61]. Utilizing the same reporter system, our lab later showed that two other SR proteins, SRSF2 and SRSF5, likely promote asTF biosynthesis in monocytes [62]. The authors also performed electromobility shift assays with the aforementioned SR proteins binding to human TF exon 5 RNA probes, indicating that a physical SR protein-RNA interaction takes place [62] (Figure 3). Interestingly, at the same time, Eisenreich et al. reported that inhibition of the PI3K/Akt pathway in human EC post TNFα stimulation resulted in the decreased production of asTF mRNA but did not alter the levels of flTF transcripts [63]. The authors also showed that the phosphorylation of SRSF1, SRSF4, and SRSF6 was increased in response to PI3K/Akt inhibition [63]. However, the interpretability of these observations is disputable. On one hand, in vitro studies of en masse SR protein activity performed in the 1990s showed that both hyper- and hypo-phosphorylation of these splicing factors promoted cell quiescence and reduced exon inclusion of splicing targets [64,65]. On the other hand, the expression levels and activity of CDC-like kinases (CLKs) and serine/arginine rich protein kinases (SRPKs) can affect the timing and specificity of SR protein residues being phosphorylated, altering their cumulative activity [66]. Therefore, given that the effects of SR protein phosphorylation vary depending on multiple factors, such as the number of phosphorylated serine and arginine residues on a given SR protein, the identity of the SR proteins being phosphorylated, and the phase of the cell cycle at the time of analysis, it is only reasonable to infer that PI3K pathway inhibition and downstream alteration of SR protein activity affected asTF biosynthesis, but the precise direction of the effect is uncertain [66].

SR proteins are relatively well-studied *cis*-acting splicing factors that bind to exonic splicing enhancer (ESE) sequences and participate in constitutive as well as alternative pre-mRNA splicing [67]. There are many other splicing regulatory proteins that our lab is investigating as possible contributors to TF pre-mRNA processing (Figure 3). For example, counterparts to ESE, there exist sequences within exons termed exonic splicing silencers, or ESS, to which heterogenous nuclear ribonucleoproteins (hnRNPs) are recruited, preventing spliceosomal assembly at a splice site [66]. In addition to ESE and ESS, there are intronic splicing enhancers and silencers that recruit SR proteins and hnRNPs, respectively, to aid with intron definition and recruitment of spliceosomal factors [66]. Finally, there are many post-transcriptional and post-translational modifications, such as acetylation, phosphorylation, and even alternative splicing of the splicing factors themselves, that influence their activity as well as their ability to interact with pre-mRNA molecules [66,68]. In sum, our “SR protein-centric” view of TF pre-mRNA processing shaped through the use of a splicing reporter construct is, at best, limited. We are currently investigating the effects of downregulating several SR proteins that participate in TF exon 5 processing and utilizing the available bioinformatics tools to identify other putative binding proteins that may regulate asTF and flTF mRNA biosynthesis. Unfortunately, large-scale, high-throughput RNA-centric approaches are yet to be optimized for the identification of all possible proteins that bind a particular pre-mRNA sequence, and analysis pipelines for these studies are still in the early stages of development.

The effects of asTF on tumor progression have been most extensively studied in BC and PDAC—two solid malignancies with strikingly different outlooks on evaluating alternative splicing aberrations that contribute to their pathobiology. While there are many studies on BC-specific alternative splicing events, there is a dearth of inquiries into the same mechanisms in PDAC [70,71,72]. Pan-cancer analysis of over 8000 tumor samples performed by the TCGA Network revealed that both BC and PDAC samples display high frequency of exon skipping [73]. Moreover, clustering the most variable splicing events in a given cancer and comparing them among cancer types revealed that PDAC and BC alternative splicing patterns were extremely similar to one another. [73]. Recent attempts to link PDAC and splicing have been limited to associative studies between patient survival and alternative splicing event frequency and type [74,75,76]. In 2019, Escobar-Hoyos and colleagues published the very first functional study of alternative splicing dysregulation in PDAC; the authors reported that PDAC cells expressing mutant p53 (one of the most common mutations in PDAC, most frequently leading to the expression of a dominant-negative protein isoform) overexpressed hnRNP kinase, which led to the expression of GTPase activating protein isoforms that further stabilized constitutively active G12D KRAS, thus facilitating cell proliferation and tumor growth [77]. The authors proposed targeting the spliceosome with specific inhibitors, which they showed to be effective; however, this approach is potentially risky due to an absolute necessity for constitutive and alternative splicing in all cells of the body, therefore raising the issue of toxicity [77]. Clearly, fundamental PDAC research is lagging in exploring the aberrations of pre-mRNA processing mechanisms, and, as such, opportunities for better diagnostic, prognostic, and treatment strategies are being missed. For example, it was recently proposed by Frankiw, Baltimore, and Li that aberrant pre-mRNA splicing in cancer cells may generate immunoreactive neoepitopes due to their escape from nonsense-mediated decay, potentially allowing for enhanced immunotherapy strategies in malignancies such as PDAC, which has, so far, been refractory to such therapeutic interventions [78]. Additionally, it may be possible to identify cancer-specific proto-oncogenic isoforms, not unlike asTF, which could be targeted without harming non-malignant cell types.

Due to the highly desmoplastic nature of PDAC lesions, cancer-cell-specific alternative splicing would likely be affected: multiple studies show that, compared to normoxia, splicing factors are differentially activated and compartmentalized in hypoxic environments [79,80,81]. For instance, CLK1 and SRPK1 are overexpressed under hypoxic conditions and alter the activity of SR proteins, hnRNPs, and other RNA-binding proteins that directly or indirectly influence alternative splicing to promote the expression of pro-survival (Bcl-xL) and pro-angiogenic (VEGFxxxa) isoforms [82]. However, we have, thus far, not detected appreciable changes in phosphorylation, expression, or localization of SR proteins in PDAC patient samples and cell lines subjected to hypoxia, while seeing only a modest elevation in CLK1 levels (K.M. and V.Y.B, unpublished observations). However, we have observed an increase in asTF and flTF levels in hypoxic PDAC cells without a change in the ratio between the two isoforms. Therefore, if the asTF/flTF ratio can be altered by influencing splicing factor activity, it is likely not due to hypoxia in the setting of PDAC. Perhaps the revelation that splicing machinery is not deregulated in PDAC to the extent seen in other solid and non-solid malignancies can reconcile the dearth of knowledge about pre-mRNA splicing in this disease. Nonetheless, it is unreasonable to assume that the most important mechanism for diversifying the cell’s proteome plays no role in the progression of PDAC—the disease that notoriously manages to harness all available means of its self-propagation. Approximately 10% of all PDAC patients carry hereditary mutations in DNA damage response (DDR) genes such as BRCA1, BRCA2, PALB2, ATM, etc. Interestingly, the same genetic alterations confer susceptibility to BC, which, as mentioned earlier, takes advantage of deregulated splicing and is most similar to PDAC in terms of its alternative splicing dynamics according to the TCGA Network analysis [73,83]. Patients with mutant DDR genes have better prognosis after chemotherapy treatment compared to those with wild-type DDR components; unfortunately, overall survival was not extended in these patients when treated with Olaparib, a PARP inhibitor for germline BRCA1/2 mutant cancers, although progression-free survival was increased [84,85]. DNA damage is also an important driver of altered pre-RNA splicing—from chemotherapy to UV radiation, both single- and double-stranded DNA breaks elicit enhanced re-localization and post-translational modification of splicing regulatory factors [68]. It is possible, therefore, that alternative splicing is only deregulated significantly in germline mutant PDAC. As such, flTF and asTF could be, in theory, transcriptionally upregulated in the spontaneous PDAC tumors, but differentially expressed due to perturbed splicing factors in familial PDAC, which opens the possibility to use such evaluations as a novel means of patient stratification in select PDAC subpopulations.

## 5. asTF and White Cell Physiology: Effect on Monocyte Recruitment and Macrophage Polarization

Tumor-associated macrophages are (TAMs) currently thought to either originate in the bone marrow and then be recruited to the tumor via chemotaxis, or to differentiate from normal tissue-derived progenitors in response to growth factors secreted by tumor cells [86]. Some experimental models of solid cancers show that IL-6, VEGFA, CSF-1, and other chemokines recruit monocytes to the tumor tissue from the bloodstream [87]. Others argue that only tissue-resident macrophages are to be found in particularly dense tumors with poor vascularization, such as PDAC tumors. However, vasculature collapse is unlikely to occur in the early stages of tumor development, and it is possible that TAMs recruited early on may contribute to further tumor development and growth [86,87]. Traditionally, in response to various stimuli, monocytes are thought to give rise to differentiated macrophages that can then be polarized across the spectrum with the following two types at its extremes: M1-polarized macrophages, which are pro-inflammatory and ‘anti-tumoral’; and M2-polarized macrophages, which can promote extracellular matrix remodeling and suppress the function of other immune cells recruited to the tumor [88]. M2-polarized macrophages are, by far, more abundant within tumors of patients with worse prognosis; however, recent evidence suggests that even pro-inflammatory signals can promote cancer progression, and ‘over-confinement’ of TAMs to these two subtypes may be unacceptably simplistic, as they appear to be far more diverse based on marker gene expression and single cell RNA sequencing data [89,90]. Nonetheless, it is clear that TAMs are associated with poor prognosis and can aid cancer cells in immune evasion, metastasis, and resistance to therapeutic agents [91].

In addition to its pro-angiogenic and pro-metastatic properties, asTF promotes monocyte adhesion to microvascular EC (MVECs) to a greater extent than flTF under various flow conditions mimicking the physiology in the bloodstream [41]. By binding to β1 integrins, asTF recruits blood monocytes through an MVEC monolayer in vitro [41]. Our group found that asTF likely plays a role in the recruitment of monocytes to the site of tissue injury site by promoting cell adhesion molecule expression on endothelial cells for leukocyte attachment, rolling, and intravasation [41]. This study is reviewed in detail by Srinivasan and Bogdanov [92]. A more recent study reported that both flTF and asTF can promote monocyte differentiation into endothelial-like cells [93]. After observing that human MVEC (cell line HMEC-1) produced and secreted high amounts of flTF and asTF after being cultured in monocyte conditioned media, Arderiu and colleagues tested whether CD16- monocytes were able to transdifferentiate into CD16+ monocytes, which are found in patients with vascular injury, in response to the TF isoforms [93]. After overexpressing asTF or flTF in MVEC, collecting the media they were cultured in, and then using it to cultivate CD16- monocytes, the authors observed an increase in CD16+ monocytes grown in either asTF- or flTF-overexpressing MVEC media [93]. These TF-isoform overexpressing HMEC-1 cells exhibited an increased attachment and assumed spindle-shaped morphology characteristic of ECs [93]. CD16+ monocytes also expressed higher levels of E-cadherin, eNOS, and vWF—well-accepted EC markers [93]. Interestingly, populations of monocytes grown in media high in either asTF or flTF also began expressing β1 integrin more frequently compared to monocytes cultured in control media, thus showing a similarity to the monocyte/macrophage trans-differentiation process via the TNFα signaling axis [93]. Both flTF and asTF evidently participate in promoting monocyte-dependent angiogenesis, with asTF playing a key role in monocyte recruitment. Despite the fact that none of the studies described above were performed in a malignant setting, it is highly plausible that asTF enhances monocyte recruitment to tumor lesions where it is expressed, and promotes neovascularization via the induction of an endothelial-like phenotype in monocytes and even cancer cells themselves. For example, we observed that BC cell-line 2A3-3, which overexpresses human asTF, forms orthotopic tumors with increased macrophage infiltrates and higher vascular density compared to empty vector and/or flTF-overexpressing 2A3-3 cells [28]. In addition, we found that inducible overexpression of human asTF in Pt45.P1 PDAC cells orthotopically injected into mice yielded increased vascularization and M2 polarized macrophage recruitment into tumors [47].

An intriguing parallel can be conceivably drawn between alternative splicing of TF and VEGF-A. Unlike *F3*, which produces only two known protein isoforms, *VEGFA* gene produces multiple structurally distinct proteins [94]. This coding plasticity allows for the induction of multiple pathways and even alterations in the receptor binding preference of certain VEGF-A ligands [94]. However, despite the difference in their numbers, both TF and VEGF-A isoforms can differentially affect angiogenesis, alter monocyte/macrophage differentiation and polarization, and directly influence the proliferative properties of cancer cells [94,95,96]. For example, similar to our findings on the effects of flTF and asTF on integrins and angiogenesis in ECs, several groups have found that VEGF-A_165_ confers pro-angiogenic properties on ECs, while VEGF_165_b impedes neovascularization [94]. Additionally, Ganta and colleagues reported that patients with peripheral artery disease have a significant positive correlation between the presence of cytotoxic M1 macrophages and VEGF_165_b and that this VEGF-A isoform directly enhances the polarization of macrophages to the M1 state, which promote tissue perfusion and healing [97]. Finally, VEGF_165_b slows down colorectal tumor growth in mice; however, it impairs the effects of the anti-VEGF-A biologic bevacizumab by binding to and sequestering it [98]. While we have not investigated the effects of intratumoral asTF on monocytes and/or endothelium in detail, we did observe an increase in F4/80+ monocytes in asTF-overexpressing tumors, and a decrease thereof after treatment with the anti-asTF monoclonal antibody RabMab1 in vivo [31,47]. The findings that support the differential involvement of VEGF-A splice isoforms both in malignancy and cardiovascular disease echo the findings regarding the differential contribution of flTF and asTF to the pathobiology of various disease states, especially if one considers the vastly different coagulant potential of the two TF forms.

## 6. asTF’s Dispensability to Normal Hemostasis: Is This All There Is to It?

In the concluding sections of this review, we discuss the potential of therapeutically targeting asTF in PDAC and other malignancies where it is overexpressed. We posit that targeting a TF isoform that is dispensable to hemostasis, yet possibly more deleterious than flTF as a cancer driver due to its solubility and autocrine effects on cancer cells, may comprise a new and promising avenue in cancer patient care. However, before we get to that, we feel it pertinent to briefly review the evidence supporting “dispensability” of asTF to such flTF-associated processes as embryonic development and blood coagulation.

The study describing human asTF for the first time reported on mildly pro-coagulant properties of this isoform—native asTF protein co-localized with platelet aggregates in artificially induced thrombi, and recombinant asTF reduced clotting times in recalcified, phospholipid-rich plasma [26]. However, a later study showed that overexpressing asTF, unlike flTF, in HEK293 cells does not increase their pro-coagulant potential [99]; a problem with that study, however, was that HEK293 were not able to secrete asTF for unclear reasons. Similarly, Boing et al. found that thrombin generation time was not significantly reduced in MIA PaCa-2 cells overexpressing—but not secreting—asTF [100]. Moreover, Unlu, Bogdanov, and Versteeg showed that, at physiological concentrations, asTF does not co-localize with flTF and is incapable of impairing flTF activity in ECs and microvesicles (MV) [101]. Lastly, because flTF-deficient mice do not survive during embryonic development, asTF was knocked into *cf3* gene locus, yielding asTF-only expressing mice that did not survive post-developmentally, and their plasma exhibited no detectable coagulation activity [102]. Overall, while highly expressed in multiple tumors and cancer cell lines, asTF does not seem to contribute to normal blood clotting to any discernible degree. Nonetheless, cofactor activity of human and murine asTF, while vastly lower than that of flTF, is not zero, and it manifests itself only in the presence of thrombogenic phospholipid [11,26,103]. We also note that MV secreted by asTF-overexpressing Pt45.P1 cells were more procoagulant compared to those secreted by parental cells, which was, at least in part, due to the increased incorporation of flTF [31]. Interestingly, the levels of flTF and asTF were increased on the surface of MV secreted by asTF-overexpressing Pt45.P1 cells, raising the possibility that both forms can, under certain conditions, contribute to thrombotic events in the circulation; thus, selective targeting of asTF may (indirectly) help ameliorate flTF-triggered thrombosis.

## 7. asTF as a Disease Biomarker

The use of clinical biomarkers has exploded in recent years. This is especially true in cancer research, where they are increasingly employed to assess disease risk, disease burden, determine the best chemotherapeutic strategy and/or response to treatment, and check for disease recurrence. Biomarkers can be grouped into several different types and measured using various means. The genotyping of tumor tissue along with immunohistochemistry (IHC) has proven to be useful in selecting appropriate therapeutic strategies. Examples of this include identifying the presence of the *BCR-ABL* fusion gene in patients with chronic myelogenous leukemia, determination of *RAS* mutation status in NSCLC and PDAC, and assessment of the presence of estrogen, progesterone, and human epidermal growth factor 2 receptors in BC. Numerous circulating proteins released by tumors (tumor markers) have also been identified, which allow the detection of hyperproliferative tissues through blood tests, thus enabling the detection of cancer; such markers can serve as a surrogate of disease relapse. Most notable among these is prostate specific antigen, which is often associated with prostate cancer, and CA19-9, a sugar moiety attached to cell surface proteins produced by the gut epithelium, which is associated with gastrointestinal cancers. Thirdly, analytes of drug metabolism allow for therapeutic drug monitoring, an important process in ensuring that patients are receiving an appropriate chemotherapeutic dosage.

As mentioned earlier, asTF circulates in the bloodstream [26]; it can be released by a variety of tissues and cell types [26,28,103]. In the majority of healthy individuals, the release of asTF into the circulation yields plasma concentrations of <100 pg/mL [32,104,105]. However, when taking into consideration all asTF biomarker studies performed thus far, it appears that ~15–30% of the “healthy population” have greater plasma asTF levels [32,104,105]. Importantly, however, it is also clear that asTF is significantly and robustly elevated in certain disease states.

Zawaski and colleagues were the first to report elevated plasma levels of asTF [104]. Reportedly, 10–20% of patients undergoing chronic hemodialysis experience thrombotic events localized to the venous access site. Zawaski et al. showed a positive correlation between plasma asTF levels and the number of thrombotic events in a cohort of these patients [104]. It was thought, at the time, that asTF could be useful as a biomarker for the prediction of such adverse events [104]. However, as the authors correctly acknowledge, this data set was likely skewed towards significance by a small number of outliers [104]. In fact, only 7/84 patients had circulating asTF levels above 200 pg/mL. In light of what we now know about the frequency of asTF levels being above 200 pg/mL in the general population, these findings are unlikely to be clinically relevant [104]. In 2010, our group conducted a pilot study on the presence of asTF in the plasma of patients with sickle cell disease (SCD); this data was presented at the 2011 Annual Meeting of the American Society of Hematology. Of the patients assessed, 14/16 had detectable levels of asTF, compared to only 2/17 healthy subjects [106]. Interestingly, this study compared adult vs. pediatric patients with SCD, finding that, in the diseased population, adults had, on average, asTF levels ~25 times higher than pediatric patients [106]. While this difference was statistically significant in the SCD population, it did not reach significance in the age-matched healthy control subjects [106]. 

In 2015 came the first report of asTF being elevated in the plasma of cancer patients, namely PDAC [32]. We reported that, on average, 15% of healthy individuals have circulating levels of asTF ≥ 200 pg/mL. This statistic was roughly the same in patients with acute coronary syndrome and diabetes mellitus. However, in patients with PDAC, 41% had asTF levels ≥ 200 pg/mL. A further distinction was found between the patients with resectable vs. non-resectable PDAC: 56% of patients with non-resectable tumors had asTF levels ≥ 200 pg/mL, compared to 22% in the resectable group. Interestingly, the circulating levels of asTF did not correlate with those of CA19-9, the only biomarker currently approved for gauging PDAC tumor burden/recurrence. Furthermore, when we applied the cutoffs for asTF ≥ 200 pg/mL and for CA19-9 ≥ 130 U/mL, we were able to predict unresectable disease in 85.7% of such patients—a figure higher than either biomarker on its own [32]. This has laid the groundwork for an ongoing prospective study assessing the levels of asTF in PDAC.

In collaboration with Angelillo-Scherrer and colleagues, we have subsequently shown, through a retrospective study, that asTF levels are also elevated in chronic liver disease (CLD) and hepatocellular carcinoma (HCC) [105]. In line with our previous studies, we found the median circulating asTF concentration to be 94 pg/mL; however, in this cohort of healthy subjects, nearly 30% had asTF levels ≥ 200 pg/mL. More than 75% of CLD and HCC patients had asTF levels ≥ 200 pg/mL. When the patients were subdivided by disease severity, median asTF levels were the highest in the low-grade liver fibrosis group and the HCC group. Interestingly, asTF levels seemed to decline as CLD progressed, before rising again once HCC developed. This became even more clear when the cirrhosis group was further subdivided by the Child–Pugh score, revealing that asTF levels rise with cirrhosis progression. In agreement with this, we found that patients with active liver disease (ALT > 50 U/L) had significantly higher levels of asTF than those with no active disease. Additionally, we were able to show that those HCC patients who received anti-cancer therapy had lower asTF levels compared to untreated patients.

Taken together, we have accomplished the initial stages of biomarker validation, showing that, in both PDAC and the CLD settings, asTF is, on average, significantly elevated. However, much work remains in qualifying asTF as a bona fide disease biomarker for its eventual use in clinical practice. The National Cancer Institute’s Early Detection Research Network has adopted the guidelines set forth by Pepe and colleagues for the five phases of biomarker development: (1) preclinical exploratory studies; (2) clinical assay validation; (3) retrospective longitudinal studies; (4) prospective screening; and (5) cancer control [107]. We have thus far completed several preclinical exploratory studies described above and developed an immunoassay (ELISA) capable of detecting plasma asTF levels in the picomolar range and above. As our ELISA setup matures, we continue to improve the assay’s precision and sensitivity, with the goal of developing an immunoassay suitable for point-of-care use. Based on our exploratory studies that featured single time-point blood draws, we surmise that 10–30% of the general population have levels of asTF ≥ 200 pg/mL; prospective, longitudinal-design studies are being planned to expand on these findings. If this degree of biological variability in the general population does hold true, it would potentially lower the usefulness of one-time assessment of plasma asTF levels in a presumably healthy individual. This, in and of itself, is not out of the ordinary, though, as many of the tumor markers of today have similar limitations. What is more likely to be important in future research is the determination of whether circulating asTF can have utility as a biomarker in certain patient subpopulations: for example, it may help guide treatment and/or predict outcomes in surgery-eligible PDAC and/or HCC; these studies are ongoing.

## 8. Therapeutic Targeting of asTF

As we mentioned earlier, it is likely preferable to target the alternatively spliced isoform of TF over the full-length isoform due to asTF’s dispensability for normal hemostasis. While there are multiple anti-flTF therapeutics currently in clinical trials, their success will depend on both efficacy and the absence of adverse (e.g., bleeding) events, which remains to be seen. As such, we continue to propose that targeting asTF is the less risky option and our in vivo findings have shown the monoclonal antibody RabMab1 to be effective at stemming the growth of orthotopic BC and PDAC tumors in mice [28,47,108]. 

RabMab1 was raised in rabbit against the stretch of amino acids in human asTF’s extreme C-terminus. It was first assessed for in vivo efficacy in an orthotopic model of BC using MDA-MB-231 cells modified to stably overexpress asTF [28]. The cells were preincubated in 2 mg/mL of RabMab1 before being injected into the inguinal fat pads of NSG mice. Following 7 weeks of tumor growth, the mice receiving injections containing RabMab1 bore tumors that were ~50% the size of those in mice receiving IgG control and/or no-treatment control injections. IHC analysis of the tumor tissue revealed a suppression of proliferation in the RabMab1 treated tumors as indicated by Ki67 staining. However, there was no difference observed in the vascular density of those tumors compared to the controls. This is most likely due to either active flTF-PAR2 signaling in the chosen cell line, or the fact that MDA-MB-231 produce heavily vascularized tumors even without asTF overexpression.

We next tested RabMab1 in a similar manner using an orthotopic PDAC model [47]. Pt45.P1 cells, a high grade PDAC cell line that produces hypovascular tumors, were preincubated with 5 mg/mL RabMab1 before being implanted into the pancreata of athymic nude mice. Following 7 weeks of tumor growth, the mice bearing tumors pretreated with RabMab1 were roughly 1/3 the size of the IgG control pretreated and/or untreated tumors. In contrast to the MDA-MB-231 tumors treated with RabMab1, the Pt45.P1 tumors treated with RabMab1 contained 50% fewer vessel structures than either control. There was also a 50% reduction in the number of infiltrating M2-polarized macrophages. Furthermore, there was a significant reduction in collagen deposition in the RabMab1 treated tumors. When we assessed the mouse plasma for circulating levels of human asTF, we found that RabMab1 pretreatment led to a >50% reduction.

Given the promising results seen with these co-incubation studies using hybridoma-derived RabMab1, we proceeded with further preclinical development and have recently reported on the humanization of RabMab1 [108]. Co-implantation experiments with a hIgG1 isotype chimeric antibody resulted in tumors that were roughly 1/4 the size of vehicle and/or IgG control tumors. Pharmacokinetic (PK) analysis of chimeric RabMab1 in C57BL/6 mice indicated a half-life of 280 h. We next proceeded to complete humanization of RabMab1 by mutagenizing specific residues in the Fab portions; this yielded an antibody with a K_D_ in the picomolar range due to an extremely low off rate. PK analysis of our humanized RabMab1 (hRabMab1) showed it to have favorable characteristics with a half-life of 908 h in C57BL/6 mice. Pharmacodynamic analysis was then performed on mice bearing preformed, orthotopic Pt45.P1 tumors. Intravenous administration of hRabMab1 every other day for 22 treatment cycles resulted in tumors that were roughly one third the size of the mice receiving vehicle or hIgG1 isotype control treatments. IHC analysis of the tumors showed that the tumors treated with hRabMab1 had fewer vascular formations, proliferating cells, and invasive M2-polarized macrophages. RNAseq was performed using RNA collected from median-size tumors in each treatment group. We found a decrease in the expression of genes associated with focal adhesion, cell motility, cell proliferation, cytoskeleton rearrangement, proteases, and cell death. Taken together, our findings demonstrate that further clinical development of hRabMab1 is likely well warranted. Future directions will take aim at assessing the effects of hRabMab1 on other primary tumors as well as metastatic spread—as a single agent, and in combination with standard of care regimens.

The landscape of TF-based diagnostics and therapeutics has been rapidly evolving over the past decade. Thus far, it appears that hRabMab1 is the only therapeutic in development that targets a specific tissue factor isoform (i.e., asTF); all others can be potentially considered as targeting “total TF”. To date, five anti-TF molecules have entered into clinical trials: MORAb-066, MRG004a, rNAPc2, ALT-836, and Tisotumab vedotin. Little is publicly known about MORAb-066 and MRG004a. NCT01761240 assessed the anti-TF mAb MORAb-066 in a phase I trial. While this trial has been completed, the results have not been published. The lack of a publication record also prevents discussion of MRG004A, an antibody–drug conjugate that has recently entered a phase I/II trial (NCT04843709). Much more is known about the remaining three molecules. Given that these molecules likely target both TF isoforms, we feel it is pertinent to briefly review these alongside hRabMab1.

rNAPc2 (recombinant nematode anticoagulant protein c2) is potent inhibitor of the TF/FVIIa complex. It forms a complex with an exosite on FX that then coordinates with TF/FVIIa to block its catalytic activity [109]. Its clinical development has focused on its anti-coagulant properties, where it was found to be well-tolerated in patients with coronary artery disease (NCT00116012) [110]. It is currently being evaluated as a COVID-19 therapeutic in a multi-center phase II/III trial (NCT04655586). rNAPc2 has also been found to have anti-tumor activity in melanoma and Lewis lung carcinoma cells, and later in colorectal cancer [111,112]. However, inquiries into its use as an anti-cancer therapeutic seem to have subsided.

ALT-836 is a monoclonal antibody directed against flTF that blocks the FX binding site [113]. As such, it, too, was initially developed as an anti-thrombotic agent. Two clinical trials have been carried out aimed at assessing the ability of ALT-836 to suppress thrombosis common in acute lung injury/acute respiratory distress syndrome (ALI/ARDS). However, in a phase I/IIa study (NCT01438853) carried out in ALI/ARDS patients, no response was observed. While no major bleeding events were reported in the study population, anemia, and hematuria were observed in some subjects [114]. ALT-836 has also been assessed in a phase I trial in combination with gemcitabine for the treatment of locally advanced or metastatic solid tumors (NCT01325558). While this study has been completed, its results have not been made publicly available. Currently, ALT-836 and its Fab fragment derivatives are being developed as TF-targeting imaging reagents in both pancreatic and triple-negative breast and anaplastic thyroid cancers. While this work remains in the preclinical stage, the ALT-836 derivates have been shown to be effective at localizing to TF-positive tumors, allowing for accurate tumor size estimation and the complete removal of subcutaneous tumors in mice (see Table 1 for references).

The most well characterized of the anti-TF therapies currently in clinical trials is Tisotumab vedotin. To date, it has been involved in nine clinical trials, six of which remain active. Tisotumab vedotin is a humanized anti-TF antibody conjugated to monomethyl auristatin E. It was selected from a panel of anti-TF antibodies due to its ability to inhibit TF:FVIIa mediated intracellular signaling, promote antibody-dependent cell-mediated cytotoxicity, and be rapidly internalized by cancer cells with only a minimal impact on coagulant potential [115]. Preclinical studies showed it to be highly effective against patient derived xenografts, even in those with low levels of TF expression, leading to its rapid introduction to the clinic. The initial first-in-human and dose-escalation studies were performed in patients with advanced or metastatic solid tumors, which included ovarian, cervical, endometrial, bladder, prostate, esophageal, lung, and squamous cell cancers of the head and neck (NCT02001623) [116]. The phase I portion of this study showed grade three dose-limiting toxicities at 2.2 mg/kg; therefore, a dosage of 2.0 mg/kg was determined to be the maximum tolerated dose for the phase II portion of the study. Of the 147 patients enrolled in the phase II portion of the study, 102 experienced epistaxis indicating there is, indeed, a high risk of major bleeding events associated with this treatment. Of the enrolled patients, 67 had a treatment-emergent serious adverse event; however, an objective response was seen in 15.6% of the patient population. These findings were deemed acceptable, and a further analysis of the cervical cancer population revealed an enhanced response to Tisotumab vedotin with a 22% objective response rate. It should be noted, however, that there was no significant association between tumor TF expression levels and clinical response to the drug [117]. A follow-up phase II trial was then conducted in patients with cervical cancer (NCT03438396): the findings were in line with those of the initial trial (24% objective response rate). Of note, there were seven complete responses in this cohort of 102 patients with an additional 17 patients showing a partial response. Once again, epistaxis was experienced by a significant number of the patient population, albeit a lower percentage—30%. Given the dearth of available second line therapeutics in aggressive cancers, despite this high risk of bleeding, Tisotumab vedotin shows much promise, which has led to its expansion into eight other trials.

Tisotumab vedotin is not the only auristatin conjugated anti-TF antibody in development; 1849-MMAE, SC1-MMAE, and 1084-MMAE have all shown anti-cancer activity against PDAC and other solid malignancies [129,130,131]. Antibody 1849 has also been conjugated to Alexa-fluor 647, where it has been utilized for imaging pancreatic and glioblastoma tumors in mice. It should be noted, however, that antibody 1849 inhibits TF-mediated coagulation [134]. Thus, the laboratory responsible for its development has turned to another anti-TF clone, 1859, for further clinical development. Antibody 1859 conjugated to epirubicin-incorporating micelles selectively accumulates in TF-expressing tumors and suppresses tumor growth in mice [135]. In addition to MMAE conjugation, antibody 1084 has also been labelled with astatine-211, an α-emitting radioisotope; 211At-anti-TF mAb was shown to have anti-tumor activity commensurate with tumor TF-expression level in gastric cancer cell-line xenografts [137]. Radio labelling has also been performed using the aforementioned ALT-836. ^131^I-ALT-836 suppressed tumor growth and prolonged the life of mice bearing orthotopic anaplastic thyroid cancer tumors. Given that ALT-836 blocks the FX binding site on flTF, bleeding risk is likely to impact its development. In theory, targeting TF outside of the FVII and FX binding pockets would not disrupt hemostasis; this has led to the development of antibody 10H10 as a potential anti-TF therapeutic. In a study of the responsiveness of glioma xenografts to anti-TF antibodies, 10H10 was as effective at prolonging mouse life as a coagulation-blocking antibody [139]. Antibody 10H10 has since been humanized in preparation for further clinical development [138]; however, recent evidence indicates that 10H10 will still have detrimental effects on coagulation due to steric interference with the assembly of the extrinsic tenase complex [39]. Thus, it seems that any antibody that binds flTF comes with the risk of disrupting coagulation. Conversely, targeting asTF is unlikely to carry any such risks, while perhaps affording superior selectivity in targeting cancer lesions. Therefore, we propose that employing hRabMab1 as a first-in-class means of targeting the TF signaling axis is of utmost interest in the preclinical realm of TF-targeting biologics.

## 9. Concluding Remarks

asTF is an interesting component of the TF system, most notably because its cell-agonist properties are exerted via non-proteolytic means. Given the long-standing focus on all things proteolytic in the TF field, the discovery and translational interrogation of asTF significantly expanded our understanding of how TF is able to modulate the properties of benign and malignant cells. We greatly anticipate new data on asTF’s role(s) in the pathogenesis of various malignancies, as well as asTF’s emerging utility as a prognostic and/or predictive biomarker and a therapeutic target.

## Figures and Tables

**Figure 1 cancers-13-04652-f001:**
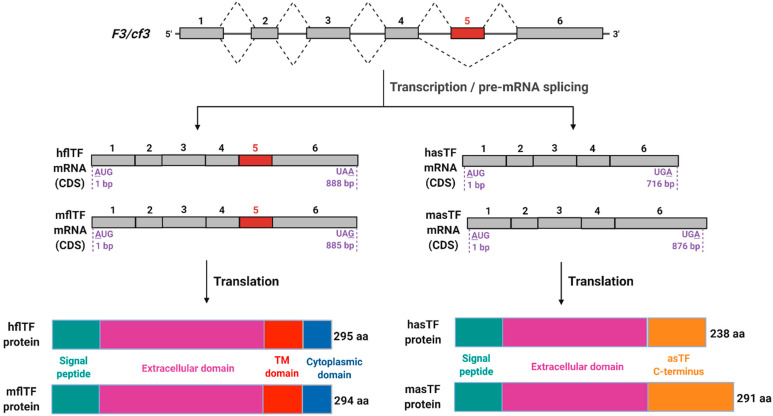
Schematic representation of *F3*/*cf3* pre-mRNA splicing products. Top to bottom: In mouse and human, *F3* and *cf3* genes, respectively, encode six exons that contribute to open reading frames; five introns are removed during pre-mRNA processing. Two mature mRNA products are produced (coding sequences (CDS) are shown) in both mouse and human; of note is the extended 3′ end of murine asTF (masTF) compared to human asTF (hasTF), which is a result of a frameshift in the coding region due to the exclusion of exon 5 during alternative splicing. Human and murine flTF proteins are of nearly identical lengths with high sequence similarity, while there is a greater variation between hasTF and masTF proteins, with the latter containing an additional 54 amino acids in its C-terminus. This figure was adapted from Bogdanov et al. (2006) and Bogdanov (2007) [11,12]. Image created with BioRender.com (accessed on 13 September 2021).

**Figure 2 cancers-13-04652-f002:**
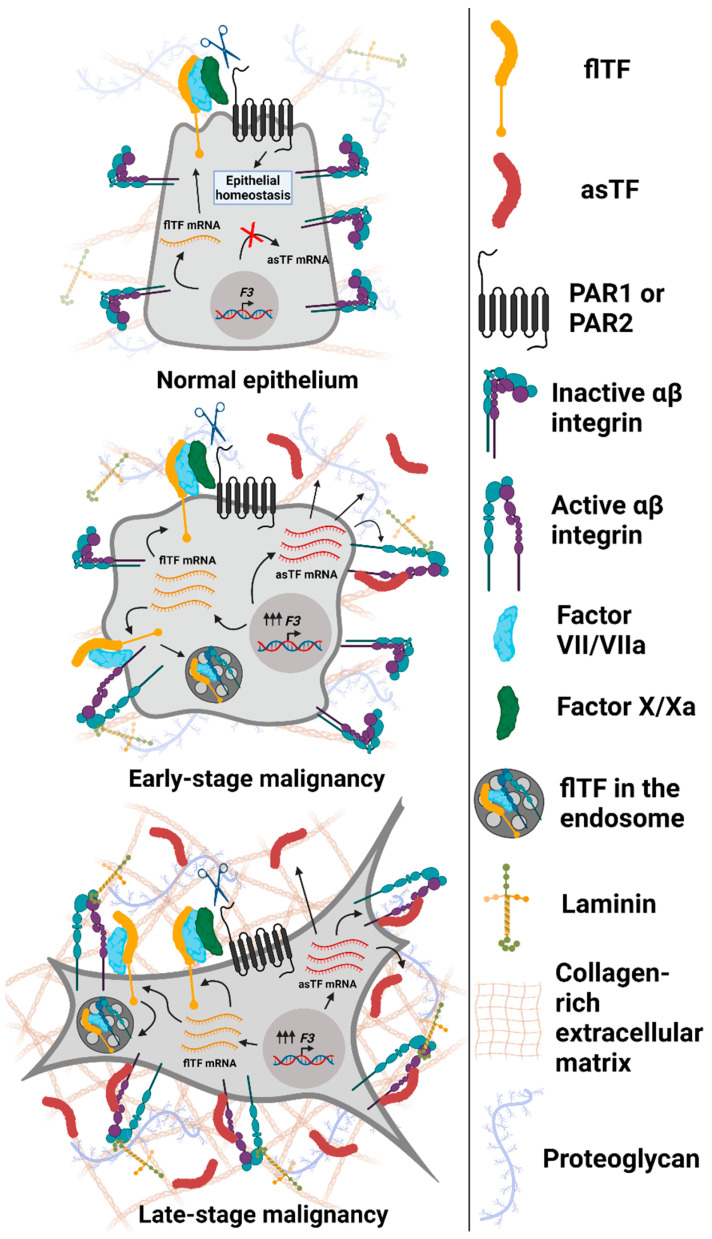
Expression and functional properties of Tissue Factor isoforms in normal and malignant epithelial cells. Top to bottom: normal epithelium (breast and/or pancreas) expresses maintenance (low) levels of flTF mRNA, which is translated into flTF protein that participates in cell/tissue homeostasis; asTF is not expressed. Early-stage transformed cells overexpress flTF as well as asTF. flTF protein activates αβ integrin subsets, promotes PAR-mediated signaling [21], and is recycled via endosomes as a means of cell-surface protein level regulation [39]. asTF activates α6β1 integrins to promote cell proliferation and migration. In addition, asTF-dependent activation of αvβ3 on endothelial cells and their subsequent migration were observed [40]. As cancer progresses, asTF protein likely accumulates in tumor stroma, leading to the increased activation of integrins, fueling cancer cell proliferation, migration, and invasion. Not shown, asTF’s properties effected via endothelial cells that promote vessel growth and monocyte recruitment [40,41]. Image created with BioRender.com (accessed on 13 September 2021).

**Figure 3 cancers-13-04652-f003:**
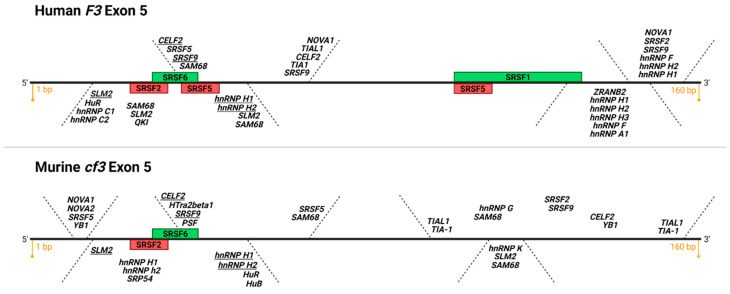
Putative and confirmed RNA binding protein motifs on human and murine exon 5 of TF. Depicted is a diagram of *F3* and *cf3* exon 5, 160 bp each, top and bottom, respectively (adapted from SpliceAid [69]). Green boxes: functionally validated SR protein binding sites that promote exon 5 inclusion (exonic splicing enhancers, ESE), red boxes: functionally validated SR protein binding sites that promote exon 5 exclusion (exonic splicing silencers, ESS). Note the conservation of functional SRSF2 ESS and SRSF6 ESE in human and mouse exon 5 (K.M. and V.Y.B., unpublished data). In italics are putative recognition consensus motifs for various classes of RNA binding proteins; consensus motifs present in human and mouse exon 5 are underlined. Image created with BioRender.com (accessed on 13 September 2021).

**Table 1 cancers-13-04652-t001:** Tissue factor-focused therapeutics developed during the past 5 years that are currently in preclinical studies, or in various stages of clinical evaluation. Cancer types are abbreviated as PDAC (pancreatic ductal adenocarcinoma), HNSCC (head and neck squamous cell carcinoma), and GAC (gastric adenocarcinoma).

Drug	Alias	Class	Conjugate	Clinicaltrials.gov Identifier	Disease	Phase	Status	References
rNAPc2	AB201	recombinant protein		NCT04655586	COVID-19	Ⅱ/Ⅲ	recruiting	[110]
				NCT00116012	Coronary disease	Ⅱ	completed	
Tisotumab vedotin	HUMax-TF, HuMax^®^-TF-ADC, TF-011-MMAE	ADC	MMAE	NCT03913741	Solid malignancies	Ⅰ/Ⅱ	Active, not recruiting	[116] [117] [118]
				NCT03245736	Solid malignancies	Ⅱ	completed	
				NCT03438396	Cervical cancer	Ⅱ	Active, not recruiting	
				NCT03485209	Solid malignancies	Ⅱ	recruiting	
				NCT02552121	Solid malignancies	Ⅰ/Ⅱ	completed	
				NCT03657043	Platinum-resistant ovarian cancer	Ⅱ	Active, not recruiting	
				NCT03786081	Cervical cancer	Ⅰ/Ⅱ	Active, not recruiting	
				NCT02001623	Solid malignancies	Ⅰ/Ⅱ	completed	
				NCT04697628	Cervical cancer	Ⅲ	recruiting	
ALT-836	TNX-832, Sunol cH36	mAb		NCT01438853	Acute lung injury/acute respiratory distress syndrome	Ⅰ/Ⅱ	completed	[114]
				NCT00879606				
				NCT01325558	Solid malignancies	Ⅰ	completed	
^64^Cu-NOTA-ALT-836		Imaging reagent	p-SCN-Bn-NOTA/ Copper-64		PDAC, triple negative breast cancer	preclinical		[119] [120] [121]
ALT-836-SERRS-NPs		Imaging reagent	SERRS-NPs		Breast cancer lung metastases	preclinical		[122]
^64^Cu-NOTA-heterodimer-ZW800		Imaging reagent	Copper-64		PDAC	preclinical		[123]
^89^Zr-DF-ALT-836		Imaging reagent	Zirconium-89		PDAC	preclinical		[124]
IRDye 800CW-ALT-836		Imaging reagent	IRDye 800CW		Anaplastic thyroid cancer	preclinical		[125]
^18^F-Fvllai		Imaging reagent	Fluorine-18		PDAC	preclinical		[126]
1849-Alexa-Fluor-647		Imaging reagent	Alexa Fluor 647		PDAC, glioblastoma multiforme	preclinical		[127] [128]
1849-MMAE		ADC	MMAE		PDAC, HNSCC, ovarian cancer, GAC	preclinical		[129] [130] [131]
SC1-MMAE		ADC	MMAE		PDAC, triple negative breast cancer, HNSCC, ovarian cancer, GAC	preclinical		[132] [130]
anti-TF Fab'-installed PIC micelles		ADC	siRNA loaded polyion complex micelles		PDAC	preclinical		[133]
anti-TF1849-NC-6300		ADC	epirubicin		PDAC	preclinical		[134]
anti-TF1859-NC-6300		ADC	epirubicin		PDAC	preclinical		[134] [135] [136]
1084-MMAE		ADC	MMAE		PDAC	preclinical		[131]
1084(^211^At-anti-TF mAb)		radioimmunotherapeutic	astatine-211		GAC	preclinical		[137]
^131^I-ALT-836		radioimmunotherapeutic	lodine-131		Anaplastic thyroid cancer	preclinical		[125]
10h10		mAb			Glioma	preclinical		[138]
hRabMab1	Rb1	mAb			PDAC, breast cancer	preclinical		[28] [47] [108]

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
