# Peer review of "Functional Characteristics and Regulated Expression of Alternatively Spliced Tissue Factor: An Update"

_cancers, 2021, doi:10.3390/cancers13184652_

Round 1
Reviewer 1 Report
On the basis of publication data by the authors and other researchers, they describe about flTF and asTF. The manuscript is well-organized and cover from alternative splicing regulation up to therapeutic strategies as anti-cancer therapy. The review is very useful for many students and researchers those want to know about the current situation of TF.
I don’t know the reason, however almost all Greek letters with integrin don’t appear properly in the PDF file of their manuscript. I hope they should be indicated correctly.
Author Response
We are grateful to Reviewer 1 for their positive opinion of our work. We apologize for the type conversion error in the names of integrins; this has been corrected.
Reviewer 2 Report
The review by Mathias et al is an interesting update on the subject of "Functional characteristics and regulated expression of alternatively spliced Tissue Factor…". The paper is well written.
It provides an opportunity to describe some aspects of the process and regulation of RNA splicing in tumor vs normal tissues.
The approach of targeting asTF looks promising for tumor treatment and may reduce side effects observed with the targeting of TF.
I have only minor comments:
The nomenclature F3 vs TF should be homogenized in the article
Since it is a review, it would be useful to have a scheme of asTF and flTF and an indication of the length (number of amino acids) of the missing sequence.
In chapter 2 "asTF is overexpressed in multiple malignancies", it would be useful to indicate how asTF is distinguished from flTF in the different articles they mention (RNAseq ?)
I am not sure that BrCa is a useful abbreviation for breast cancer
There is a problem of PDF conversion, presumably for symbol characters which are lost upon conversion rendering the identification of integrins sometimes difficult. Surely, I don't know if the authors are responsible for this.
In the legend of fig 1, it would be interesting to mention the integrins that are activated by asTF
In the sentence lines 426-432 "…..PDAC cells expressing mutant p53 (one of the most common mutations in PDAC, most frequently leading to the expression of a dominant-negative protein isoform) overexpressed hnRNP kinase, which led to the expression of KRAS-inactivating proteins, thus enabling increased cell proliferation and tumor growth [73].", I am surprised by the terms "KRAS-inactivating proteins", how could that lead to increased cell proliferation and tumor growth ?
In chapter 7: "asTF as a disease biomarker", line 685, it is said that 10-30% of the normal population have more than 200pg/ml in their plasma. Did the authors observe a longitudinal variation of the concentration in the normal and diseased subjects ?
In chapter 3, when the RabMab1 is introduced, the authors should explain how this Mab recognizes asTF and not flTF. Is the epitope identified ? Is it possible that there is a conformational change of the EC region when the TM region is missing ? Other explanations ?
Aurstatin lines 791, 818: do the authors mean Auristatin ?
I am not sure that the part of Chapter 8 dedicated to the targeting of TF in general and not specifically asTF is not very relevant for the subject of the article. Maybe the listing of trials in table 1 with different molecules targeting TF could be moved to Suppl materials.
Author Response
We are grateful to Reviewer 2 for their thorough evaluation of our manuscript, and such a positive view of our work. Our point-by-point responses are below.
The nomenclature F3 vs TF should be homogenized in the article
Response: Done as suggested.
Since it is a review, it would be useful to have a scheme of asTF and flTF and an indication of the length (number of amino acids) of the missing sequence.
Response: Done as suggested (please see newly added fig.1).
In chapter 2 "asTF is overexpressed in multiple malignancies", it would be useful to indicate how asTF is distinguished from flTF in the different articles they mention (RNAseq ?)
Response: done as suggested.
I am not sure that BrCa is a useful abbreviation for breast cancer
Response: we changed "BrCa" to "BC" throughout the paper.
There is a problem of PDF conversion, presumably for symbol characters which are lost upon conversion rendering the identification of integrins sometimes difficult. Surely, I don't know if the authors are responsible for this.
Response: we apologize for having overlooked the font issue; this has been corrected.
In the legend of fig 1, it would be interesting to mention the integrins that are activated by asTF
Response: done as suggested.
In the sentence lines 426-432 "…..PDAC cells expressing mutant p53 (one of the most common mutations in PDAC, most frequently leading to the expression of a dominant-negative protein isoform) overexpressed hnRNP kinase, which led to the expression of KRAS-inactivating proteins, thus enabling increased cell proliferation and tumor growth [73].", I am surprised by the terms "KRAS-inactivating proteins", how could that lead to increased cell proliferation and tumor growth ?
Response: thank you for pointing this out; we edited this sentence to better clarify the mechanism.
In chapter 7: "asTF as a disease biomarker", line 685, it is said that 10-30% of the normal population have more than 200pg/ml in their plasma. Did the authors observe a longitudinal variation of the concentration in the normal and diseased subjects ?
Response: this part was edited to clarify that the published observations were based on studies involving single-timepoint blood draws, and that the longitudinal studies are currently ongoing.
In chapter 3, when the RabMab1 is introduced, the authors should explain how this Mab recognizes asTF and not flTF. Is the epitope identified ? Is it possible that there is a conformational change of the EC region when the TM region is missing ? Other explanations ?
Response: we clarified the epitope that was used to generate this human asTF-specific antibody. We do not have definitive data regarding the conformational changes that may indeed occur in the EC portion due to its shortening, and/or due the presence of a unique c-terminal domain.
Aurstatin lines 791, 818: do the authors mean Auristatin ?
Response: thank you for pointing this out; typos were corrected.
I am not sure that the part of Chapter 8 dedicated to the targeting of TF in general and not specifically asTF is not very relevant for the subject of the article. Maybe the listing of trials in table 1 with different molecules targeting TF could be moved to Suppl materials.
Response: we added new language to the corresponding section to explain to the reader that the predominant majority of these TF-targeting entities are likely to target both isoforms of TF; for this reason, we prefer to have the table to remain in the body of the text to put a better perspective on how / where hRabMab1 fits into this current landscape; we hope the Reviewer agrees. The table has been modified to be more reader-friendly, and now fits into a single page.